# Fabrication of Titanium and Copper-Coated Diamond/Copper Composites via Selective Laser Melting

**DOI:** 10.3390/mi13050724

**Published:** 2022-04-30

**Authors:** Lu Zhang, Yan Li, Simeng Li, Ping Gong, Qiaoyu Chen, Haoze Geng, Minxi Sun, Qinglei Sun, Liang Hao

**Affiliations:** 1Gemmological Institute, China University of Geosciences, Wuhan 430074, China; luzhang18468235771@163.com (L.Z.); yanli@cug.edu.cn (Y.L.); simengli@cug.edu.cn (S.L.); 2201710163@cug.edu.cn (P.G.); jojo_chen128@163.com (Q.C.); 13194594977@sina.cn (H.G.); smx8632@163.com (M.S.); 2Hubei Gem & Jewelry Engineering Technology Research Center, Wuhan 430074, China

**Keywords:** selective laser melting, thermal management materials, titanium-coated diamond/copper composites, copper-coated diamond/copper composites

## Abstract

The poor wettability and weak interfacial bonding of diamond/copper composites are due to the incompatibility between diamond and copper which are inorganic nonmetallic and metallic material, respectively, which limit their further application in next-generation heat management materials. Coating copper and titanium on the diamond particle surface could effectively modify and improve the wettability of the diamond/copper interface via electroless plating and evaporation methods, respectively. Here, these dense and complex composites were successfully three-dimensionally printed via selective laser melting. A high thermal conductivity (TC, 336 W/mK) was produced by 3D printing 1 vol.% copper-coated diamond/copper mixed powders at an energy density of 300 J/mm^3^ (laser power = 180 W and scanning rate = 200 mm/s). 1 and 3 vol.% copper-coated diamond/copper composites had lower coefficients of thermal expansions and higher TCs. They also had stronger bending strengths than the corresponding titanium-coated diamond/copper composites. The interface between copper matrix and diamond reinforcement was well bonded, and there was no cracking in the 1 vol.% copper-coated diamond/copper composite sample. The optimization of the printing parameters and strategy herein is beneficial to develop new approaches for the further construction of a wider range of micro-sized diamond particles reinforced metal matrix composites.

## 1. Introduction

Metal matrix composites (MMCs) are one of the most advanced potential materials in thermal management applications. The addition of nano/micro-sized fillers into metal matrix can reinforce the mechanical, thermal, and electrical performance of MMCs. To date, only relatively simple MMC designs with limited functionality have been produced by various solid (e.g., powder metallurgy) and liquid (e.g., stirring and squeeze casting) methods [1].

Diamond/copper composites are the excellent examples of MMCs. Diamond is a next-generation thermal management material, and the density of these composites is below that of pure copper. Many researchers employed diamond particles to tailor copper’s coefficient of thermal expansion (CTE) and enhance the thermal conductivity (TC) [2]. Although these diamond-particle-reinforced MMCs have great prospects in heat dissipation, it’s not easy to fabricate complex structure due to the complexity of processing diamond-based materials. With high energy density, small laser spot, and high cooling rate, selective laser melting (SLM) is a unique additive manufacturing (AM) technology that is used to construct complex designs [3], which has drawn the attention from industry and academia [4]. The method adopts a high-energy laser beam for melting metal powders layer by layer in 3D computer aided design (CAD) data [5,6], after which the micro-melted metal pool cools to form precise metal parts [7,8,9,10,11]. Thus, combining SLM’s ability to create complex designs with an MMC’s excellent performance could amplify the capabilities and reshape the application [12,13].

However, diamond and copper have extremely poor wettability, causing weak interfacial bonding and large interfacial thermal resistance [14,15]. There are two general solutions applied for this issue [16]. The first is matrix alloying, which leads the alloying element, for example, titanium [15], boron [17,18], zirconium [19,20], or chromium [21], into the copper matrix. However, it is difficult to precisely control the additive amount, thus resulting in excess elements residual in matrix, thereby decreasing the TC of MMC. The second is surface modification, which uses strong carbide-forming elements, like titanium [22], chromium [23,24], tungsten [25], or molybdenum [26], onto the diamond particle surface. The abovementioned four metal layers can react with diamond and partly dissolve in a copper matrix. Since the formation free energy of titanium carbide (TiC) is sufficiently low, it can easily form carbide in a short time and chemically bond with diamond. The interface layer formed serves as a bridge connecting copper and diamond, thereby increasing the composite’s TC. Sun et al. [27] reported that the copper/diamond composites with titanium-coated diamond particle composites were synthesized by mechanical alloying, after annealing at 800 °C, the TC of a 40 vol.% diamond/copper composite reached 409 W/mK. The titanium-coating combination to the surface metallization of diamond particles helps to improve the copper/diamond interface and obtains a high TC within the composite material. Zhang et al. [28] reported that copper coated diamond powder in a variable mass ratio of diamond and copper was fabricated through an electroless plating process, before the coated powder was sintered through spark plasma sintering, the TC of a 70 vol.% diamond/copper composite reached 404 W/mK. In conclusion, they all have these disadvantages of the single sample shape and the relatively terrible interface bonding.

This research investigates the comparison for the thermal and mechanical performance between copper-coated diamond/copper combined materials manufactured by SLM and those with titanium-coated diamond/copper combined materials. In the current research, dense complex diamond/copper combined materials with titanium and copper-coated diamond particles were successfully 3D printed by implementing different forming processes and 3D printing strategies. A series of experiments were carried out via SLM single-line scanning to determine the appropriate processing parameters, and a cubic sample was prepared to characterize the composites’ morphology and microstructure. The optimization of 3D printing parameters and strategy herein can be useful to develop new approaches for the further construction of a wider range of diamond-particle-reinforced MMCs [29]. In this work, the dense 1 vol.% copper-coated diamond/copper composite manufactured via SLM displays good interfacial bonding of the copper matrix and diamond reinforcement, and excellent thermal and mechanical performance were firstly revealed, which would serve as a promising candidate for thermal management material.

## 2. Materials and Methods

### 2.1. Materials

The micron-level pure gas atomized copper powders were purchased from China Metallurgical Research Institute (purity of 99.99%), the particles of which were primarily spherical, allowing them to improve the fluidity and bulk density in comparison with polyhedral particles [30]. The average powder particle size of 18.856 ± 15 μm was below that of the laser spot (30 μm). The diamond particles were purchased from Henan Yuxing Sino-crystal Micro-diamond Co., Ltd., with an average size of approximately 25 μm.

In this experiment, copper and titanium were directly deposited on the diamond particle surface via electroless plating (Figure 1a) and evaporation methods (Figure 1b), respectively. Table 1 lists the mix ratios of copper powder and coated diamond particles, combined in a ball mill at 100 rpm for 3 h, before which were dried for 3 h at 60 °C and then sifted through a 400 mesh.

### 2.2. Preparation of the Titanium and Copper-Coated Diamond/Copper Components through SLM

In order to obtain the high-quality titanium and copper-coated diamond/copper samples, the SISMA MYSINT100 system with a neodymium-doped yttrium aluminum garnet fiber laser (wavelength: 1060 nm; maximal output laser power, 180 W; laser spot size, 30 μm) was used to investigate the detailed parameters from singlet to cubic formation in high-purity N_2_ atmosphere (residual oxygen content < 0.5 vol.%). The titanium and copper-coated diamond/copper powders were protected from oxidation. Rectangular contour (1 × 3 mm^2^) and cubic (5 × 5 × 5 mm^3^) samples were used for the monorail and block experiments, separately (Figure 1c). The chessboard laser scanning strategy was used for the cubic samples, each layer was separated into four squares, and the scanning direction in each square was perpendicular to the adjacent square. The no-hatch laser scanning strategy was used for the rectangular contour samples. A single-line scan was performed via SLM to determine the appropriate processing parameters, which were then used to prepare the cubic samples for the morphology and microstructure characterization.

The hatch distance indicated the degree to which the titanium and copper-coated diamond/copper powders were repeatedly scanned by the laser. At the same laser power and scanning rate, the smaller the hatch distance, the greater the laser’s influence on the composites. A variety of volumetric laser energy density (*D*) was adopted to determine the finished part parameters. *D* is computed through Equation (1):(1)$$D=\frac{P}{h\times t\times v}$$
in which *P*, *v*, *h* and *t* refer to the laser power, scanning rate, hatch distance and layer thickness, respectively.

### 2.3. Characterization Techniques

The surface morphology and microstructure were observed via optical microscopy (OM, Leica A205, Wetzlar, Germany) and scan electron microscopy with energy dispersive spectroscopy (SEM-EDS, Hitachi-Su8010, Hitachi High-Tech, Clarksburg, MA, USA). Prior to the microscopic observation, the specimens were polished and etched (10 s within a mixed solution of 100 mL distilled water, 5 mL HCl and 5 g FeCl_3_). The width of a single weld pool within the printed composites was assessed via image analysis with ImageJ. Titanium-coating and copper-coating thickness on the diamond particle surface were determined through focused ion beam (FIB) and SEM. The coated diamond cross-section specimens were prepared using FIB milling. X-ray diffraction (XRD, Bruker D8 Advance, Rheinstetten, Germany) employing Cu Kα radiation at 40 KV and 40 mA within the scope of 2*θ* = 5°–90° was used to test the phase structure with a step size of 0.02°. The composite density was determined adopting Archimedes’ law. A Netzsch LFA 427 Transient Laser Flash machine and the calorimetric technology were used for measuring the thermal diffusivity and specific heat, respectively. The TC was obtained by multiplying the composite density, thermal diffusivity, and specific heat. A carbon spray was used to coat the top and bottom surfaces of the specimens (Ø 12.7 × 3 mm) with graphite to improve their capacity for absorbing the applied energy. The mean TC of each specimen was determined by measuring three parallel positions. Finally, samples (3 × 4 × 25 mm^3^) were used to test the bending strength and CTE. The bending strength test was performed applying an Instron 5569 Universal Testing machine, and a dilatometer (DIL 402C, Netzsch, Selb, Germany) was used to examine the CTE of the composite materials from room temperature to 400 °C. The roughness of the top surface was measured by a laser scanning confocal microscope (OLYMPUS OLS4100) with Gaussian filtering to evaluate the quality of 1 vol.% copper-coated diamond/copper combined materials surfaces, the OLS4100 can correctly identify a measuring position and easily perform rough-ness measurement of a target micro area, the accuracy of height measurement was less than 0.2μm error per 100μm.

## 3. Results and Discussion

### 3.1. Characterization and Analysis

The vacuum evaporation technique was used to coat the diamond particle surface with titanium [31]. Figure 2a displays the titanium-rich regions on the diamond sites via EDS element mapping, showing the preference of titanium for diffusion bonding with the diamond. The EDS elemental map of the titanium-coating layer is exhibited in Figure 2b, demonstrating the enrichment of Ti. The protective deposition on the surface edge caused the Pt signals. The titanium layer thickness on the diamond particle surface ranged between 93.04 and 122.8 nm. The coating layer was closely attached to the diamond particles, and the clear stepped surfaces illustrated that the titanium and diamond were strongly chemically bonded. Diamond particles were coated employing copper through electroless plating for enhancing its wettability with molten copper, the copper-coating layer was thick enough to melt with the copper powder that only copper region on the diamond site was displayed by EDS elements mapping (Figure 2c). And the copper layer thickness on the diamond particle surface ranged between 0.99 and 1.77 μm (Figure 2d).

The series of diffraction peaks marked in black plum blossom were indexed to the TiC (PDF#32-1383), and the peaks with rhombus were indexed to the diamond (PDF#06-0675). No impurities were characterized by the XRD (Figure 2e). As the natural non-wettability of diamond and copper made it difficult to form intermediate products and impossible to achieve perfect interface bonding [8], active element titanium was coated on the diamond particle surface for clearly generating carbide TiC at the interface. As a result, the metal matrix covered the diamond surface well, improving the poor interface bonding of the diamond and metal. Existing research confirmed these phenomena [15,32,33,34,35,36]. The series of diffraction peaks marked with heart symbols were indexed to copper (PDF#04-0836), and those peaks with rhombus were indexed to diamond (PDF#06-0675). No impurities were characterized by the XRD (Figure 2f). The copper plating on the diamond particle surface improved the poor interface adhesion between the diamond and copper.

### 3.2. Formation of the SLM Titanium and Copper-Coated Diamond/Copper Composites

#### 3.2.1. SLM Manufacturing of Titanium-Coated Diamond/Copper Composites

The rectangular contour samples were printed in the single-track experiment, and their morphology was characterized and correlated with the process parameters. Due to scanning a single-track layer-by-layer along the Z-axis to form each rectangular contour sample, the width of a single wall was the width of a single weld pool. Samples with different process parameters were observed via OM to estimate the range of process parameters in the cubic experiment. The SLM laser parameters selected for the single-track formation test of the titanium-coated diamond/copper combined materials were as follows. The laser power was improved from 140 to 160 and 180 W, and the scanning rate was improved from 200 to 300 and 400 mm/s at a fixed powder layer thickness of 0.025 mm. It was important to observe the continuity of the molten pool, as any discontinuity would produce defects, for example, porosity, delamination and surface roughness. Figure 3a,b exhibited that the superior process parameters of the rectangular contour samples with 3 and 5 vol.% titanium-coated diamond/copper composites comprised a laser power of 140 W and a scanning rate of 200 mm/s, that with 1 vol.% titanium-coated diamond/copper composites comprised a laser power of 180 W and a scanning rate of 200 mm/s. In the single-track experiment, when the layer thickness was below the median powder particle size (0.025 mm), a significant amount of powder particles larger than the layer thickness would rub the previous layer in the melting region as the scraper moved, leading to failure of the print formation. Therefore, a layer thickness of 0.025 mm was optimal.

Compared with the single-track experiment involving only three process parameters (layer thickness, scanning rate and laser power), the cubic experiment further considered the hatch distance and the scanning strategy. The hatch distance (*h*) was calculated as follows:(2)h=(1−Hr)×w
where *Hr* and *w* indicate the overlap rate and melt pool width, respectively. The higher the overlap rate, the higher the density and the lower the porosity. The highest densities and lowest porosities were obtained for each parameter group when the melt pool overlap rate reached 60%. The center of the next scan track overlapped the edge of the adjacent single-track melt pool and was partially remelted to achieve good metallurgical bonding between the pools and reduce the porosity. However, when the overlap was greater than 60%, the porosity increased slightly. If the laser energy density increased further upon a rise in the laser power or a decrease in the hatch distance, the porosity increased slightly when the density reached its peak value, that is, when the porosity was at its lowest. Existing research confirmed these phenomena [37,38,39].

The data of the single-track formation test of the 1, 3 and 5 vol.% titanium-coated diamond/copper combined materials could be obtained through the ImageJ software, the melt pool width was found to be approximately 250 μm, and the h was calculated as approximately 100 μm. Figure 3c shows the unpolished samples of the SLM-printed 1, 3 and 5 vol.% titanium-coated diamond/copper combined materials. The 1 and 3 vol.% composites had higher relative density than that of the 5 vol.% composite, the relative density of 1, 3 and 5 vol.% titanium-coated diamond/copper composites were 96%, 90% and 81%, respectively. A higher relative density is related to better specimen performance [40]. Hence, 1 and 3 vol.% coated diamond/copper composites featuring high relative density were chosen for the investigation below.

Additionally, with the emergence of a metal vapor above the molten pool, a recoil pressure was induced onto the surface of pool. Meanwhile with the quick development of a thermal gradient inside the liquid, a flow of molten metal was produced from the hot region to the coldest one, referred to as the Marangoni role. Both roles brought particle ejection, known as recoil-induced ejection, as described in Figure 4 [41,42]. Besides, a denudation area was built on the sides of the molten track, where we could sweep powders under the flow of the metal vapor and the gas shield. This kind of particle ejections was known as entrainment particle (Figure 4) [43]. Metal vapor-induced entrainment caused powder-spattering behavior [44]. To be intuitive, micrometric particles changed the surface tension, molten pool rheology, and metal vapor density according to declaration [45,46,47]. Also, it was believed that the imbalance of the molten pool, hot spatter collision during flight and entrained particles gathered on the solidified area were among the major driving forces for big spatter production. A pronounced spattering situation was found while mixing 5 vol.% coated diamond with copper by comparing with a pure copper bed condition, thus determining the negative role of the coated diamond in the molten pool. The hot particles partially fused with the solid materials after falling back into the powder bed, cooled down after ejection [29].

#### 3.2.2. The Formation of SLM Copper-Coated Diamond/Copper Composites

The preparation of dense copper/diamond composite materials with a high TC required the shaping of a strong interphase layer between the diamond and copper due to their incompatibility [48,49]. For improving the wettability of metal matrix and diamond, many researchers adopted electroless plating for coating the copper layer on diamond powders [50]. The superior SLM process parameters of the rectangular contour samples with 1 vol.% copper-coated diamond/copper composites comprised a laser power of 180 W and a scanning rate of 200 mm/s (Figure 5a), that with 3 vol.% copper-coated diamond/copper composites comprised a laser power of 160 W and a scanning rate of 100 mm/s (Figure 5c,d). According to the quality of the melted rails, the processing window was separated into four different zones (Figure 5d,e), that is, a weak sintering zone (52.8%; A), an unstable melting zone (22.2%; B), a continuous track zone (2.8%; C), and an over-melting zone (22.2%; D), in reference to their different line-energy densities (LEDs, J/m). The less continuous track area suggested a formation difficulty under the constrained process parameters. It was not easy to form a scanning track due to the insufficient LED within zone A. The powder failed to form a stable width track within zone B, and a large amount of unmelted powder adhered to the surface due to insufficient energy. In zone C, the melt flow in the molten pool stabilized and had a sufficient penetration depth into the previous layer, thus obtaining a relatively smooth trajectory when the input LED was 1600 J/m. When the LED was too high (>1700 J/m) in zone D, a track featuring a width of approximately 517 μm occurred, and micro-cracks appeared due to the accumulation of excess heat related to the high power and low scanning rate caused by the high residual stress [51].

The printing parameters were optimized to form the dense copper/diamond composites. Scanning speeds ranging from 50 to 300 mm/s and laser power levels ranging between 130 and 180 W caused numerous printing defects (e.g., balling and pores). To obtain parts with minimal printing defects, the narrow processing window was determined, with a scanning rate of 100 mm/s and a high laser power (160 W). With a high laser power (170–180 W) and a low scanning rate (50 mm/s), the printed parts exhibited superfusion (Figure 5c,e). These parameters increased the molten pool size, thereby increasing the height and width of the powder tracks. 

Additionally, the values of roughness gradually decreased with increasing laser power (Figure 6). When the laser power reached the maximum value of 180 W, the surface roughness value *Sa* reached a minimum value of 5.751 μm at the scanning rate of 200 mm/s. While the scanning rate was varied between 50 and 200 mm/s, the values of roughness gradually decreased with increasing scanning rate. When the scanning rate was 200 mm/s, the surface roughness value *Sa* reached a minimum. While the scanning rate was varied between 200 and 300 mm/s, the values of roughness gradually increased with increasing scanning rate. Laser power and scanning rate played important roles in the roughness of the composite. The lifetime of the molten pool is the key parameter that influence the flatness and surface roughness, which will be discussed in the future [52].

### 3.3. Comparison of the SLM Titanium and Copper-Coated Diamond/Copper Composites

Interfacial bonding holds the key to determining the thermal and mechanical performance of composites [53]. 1 vol.% copper-coated diamond/copper composite sample showed relatively better interface bonding of the copper matrix and diamond reinforcement (Figure 7c), no remarkable defects like flaws or cracks were seen at the interface. The pull-out of diamond particle was merely discovered in the polished surface, implying strong interfacial bonding between the copper matrix and diamond particles. However, 1 vol.% titanium-coated diamond/copper combined material sample showed the pull-out of diamond particle that could be discovered in the polished surface (Figure 7a). For the 3 vol.% titanium-coated diamond/copper combined material sample, the copper matrix and diamond reinforcement bonding were extremely poor and showed obvious cracking (Figure 7b). Resulting from poor interface bonding between diamond and copper, 3 vol.% titanium-coated diamond/copper composite showed a low TC of 57 W/mK at an energy density of 280 J/mm^3^ (140 W, 200 mm/s). There were small cracks in the edge area between copper matrix and diamond reinforcement for the 3 vol.% copper-coated diamond/copper composite sample (Figure 7d). Good interfacial bonding was exhibited in the 3 vol.% copper-coated diamond/copper composite sample for comparison with the 3 vol.% titanium-coated diamond/copper composite sample. The reason why the interfacial bonding of titanium-coated diamond/copper composite was worse than that of copper-coated diamond/copper composite was that introducing electroless copper plating process could avoid essentially the particles gathering and it could improve the relative density, the interfacial bonding and the TC of the diamond/copper composites [28].

The bending strength of 1 and 3 vol.% copper-coated diamond/copper composites significantly exceeded that of corresponding titanium-coated diamond/copper composites. The maximum bending strength of 3 vol.% copper-coated combined materials was 108 MPa, while that of the 3 vol.% titanium-coated combined materials was only 36 MPa. The maximum bending strength of the 1 vol.% copper-coated combined materials was 150 MPa, that of the 1 vol.% titanium-coated composites was 148 MPa. And the printed composites with 1 vol.% copper-coated diamond were approximately three times stronger than the printed copper (≤58 MPa) for the bending strength. When the diamond concentration rose from 1 to 3 vol.%, the bending strength decreased as the relative density decreased (Figure 7e), the viscosity of melt increases obviously as the diamond particle content increases at a constant size, resulting in the decrease in the fluidity and deterioration in the sample surface quality, leading to the decrease in relative density [54]. These experiments indicated that a moderate TC (174 W/mK) was produced by printing 1 vol.% titanium-coated diamond/copper mixed powders at an energy density of 360 J/mm^3^ (180 W, 200 mm/s), a maximum TC (336 W/mK) was produced by printing 1 vol.% copper-coated diamond/copper mixed powders at an energy density of 300 J/mm^3^ (180 W, 200 mm/s) and a moderate TC (162 W/mK) was produced by printing 3 vol.% copper-coated diamond/copper mixed powders at an energy density of 533 J/mm^3^ (160 W, 100 mm/s). Additionally, the printed composites with 3 vol.% copper-coated diamond were approximately three times larger than those with 3 vol.% titanium-coated diamond for the TC. A moderate TC (183 W/mK) was produced by printing the copper powders at an energy density of 171 J/mm^3^ (180 W, 350 mm/s) as shown in Figure 5b. This difference was likely caused by the dissimilarity of those interfacial bonding between the copper matrix and diamond reinforcement. The higher energy density required to print diamond/copper composites compared to pure copper was caused by solid coated diamond particles into the molten copper pool. The solid coated diamond particles improved the molten metal viscosity and restricted its ability to flow and fuse. And as the coated diamond particle content further increased, the porosity and the laser absorptivity of the mixed powder further increased, the powder fluidity and the thermal conductivity further decreased [6].

Figure 7f shows the CTE curves of the 1 and 3 vol.% titanium and copper-coated diamond/copper composites, as well as copper upon raising the temperature from 30 to 400 °C. The CTE of 1 and 3 vol.% copper-coated diamond/copper composites was significantly below that of corresponding titanium-coated diamond/copper composites, the minimum CTE of 1 vol.% copper-coated diamond/copper combined materials was very close to that of the copper. The shear stress had little effect on the interior of the copper-coated diamond/copper composite during heating, demonstrating the interfacial bonding of the copper matrix and diamond was better in the 1 vol.% copper-coated diamond/copper composites. During the heating process, the shear stress along the interface of the copper matrix and diamond had little effect on the CTE and led to a better thermal stability. This highlighted the advantages of copper plating on diamond particle surfaces, while the copper-coated diamond/copper composite properties reached high levels in comparison.

## 4. Conclusions

SLM technology was used to form titanium and copper-coated diamond/copper composites. The microstructure, roughness, interface bonding, thermal and mechanical performance were studied.

(1) The values of roughness gradually decreased with increasing laser power. When the laser power reached the maximum value of 180 W, the surface roughness (*Sa*) reached a minimum value of 5.751 μm. The surface roughness *Sa* reached a minimum at the scanning rate of 200 mm/s.

(2) 1 vol.% copper-coated diamond/copper composite sample showed relatively best interface bonding of the copper matrix and diamond reinforcement, corresponding the lowest CTE and the strongest bending strength.

(3) 1 vol.% copper-coated diamond/copper composites had the highest TC (336 W/mK) at an energy density of 300 J/mm^3^ (180 W, 200 mm/s). 3 vol.% copper-coated diamond/copper composites had the moderate TC (162 W/mK, 533 J/mm^3^, 160 W, 100 mm/s). 1 vol.% titanium-coated diamond/copper composites had the moderate TC (174 W/mK, 360 J/mm^3^, 180 W, 200 mm/s). 3 vol.% titanium-coated diamond/copper composites had the lowest TC (57 W/mK, 280 J/mm^3^, 140 W, 200 mm/s). The copper powders had the moderate TC (183 W/mK, 171 J/mm^3^, 180 W, 350 mm/s).

The article offered electroless plating and evaporation methods for SLM to coat copper and titanium on the diamond particle surface for modifying and improving the wettability of diamond/copper interface, which opened up a new way for laser 3D printing technology to print a broad range of diamond-particle-reinforced MMCs. Thereby, unleashing their full potential for electronic package and thermal management applications.

## Figures and Tables

**Figure 1 micromachines-13-00724-f001:**
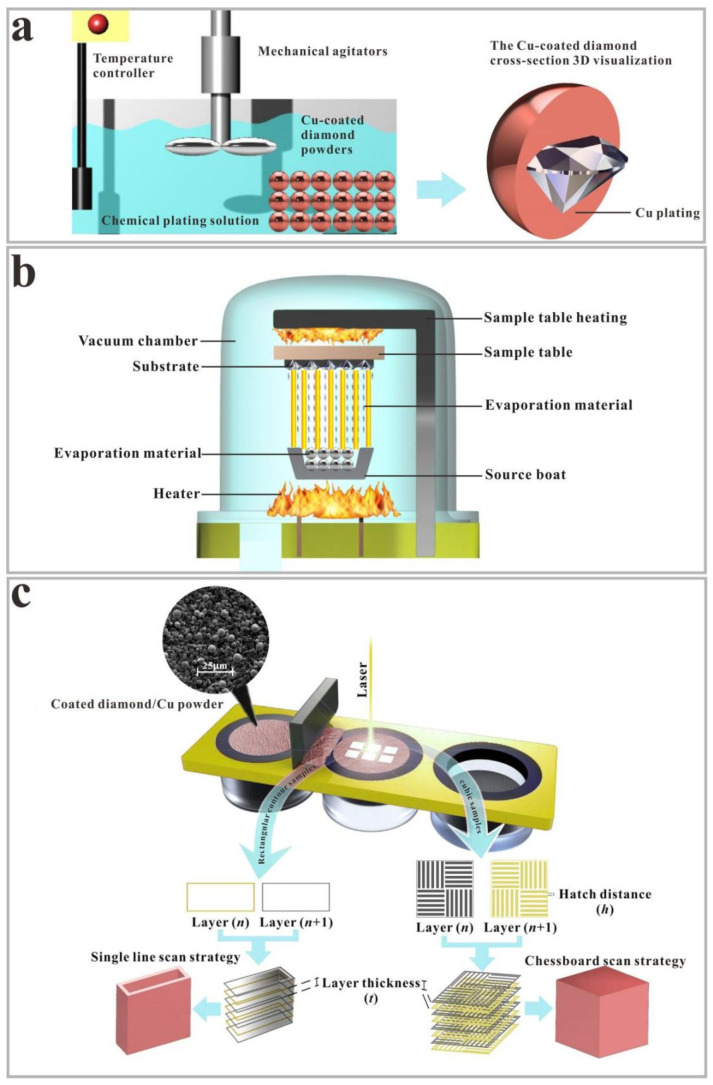
Schematic diagram of depositing copper and titanium on the diamond particle surface via (**a**) electroless plating and (**b**) evaporation process, respectively; (**c**) The preparation process for rectangular contour and cubic samples.

**Figure 2 micromachines-13-00724-f002:**
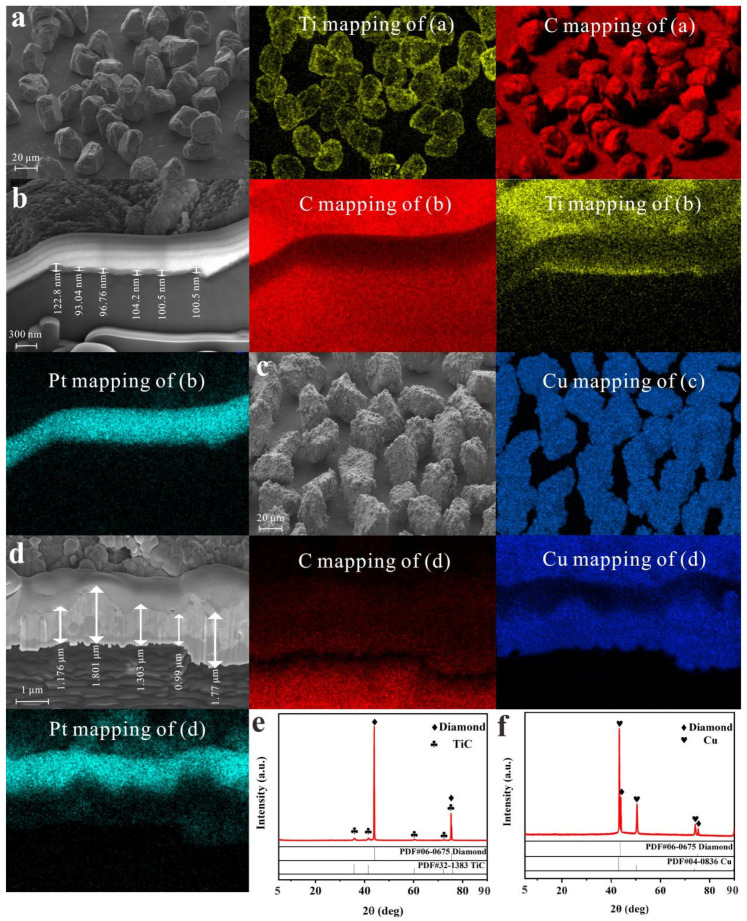
(**a**) The morphologies of the titanium-coated diamond particles and the corresponding EDS element mappings; (**b**) The thickness of the titanium-coating layer on the diamond particle surface and the corresponding EDS element mappings; (**c**) The morphologies of the copper-coated diamond particles and the corresponding EDS element mappings; (**d**) The thickness of the copper-coating layer on the diamond particle surface and the corresponding EDS element mappings; The XRD pattern of (**e**) the titanium-coated and (**f**) the copper-coated diamond particles.

**Figure 3 micromachines-13-00724-f003:**
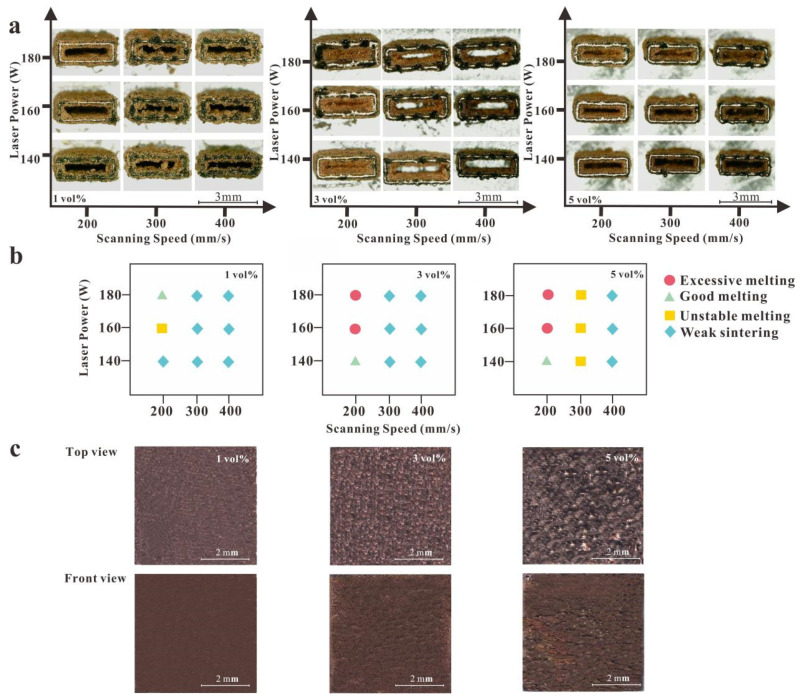
(**a**) Rectangular contour samples of the 1, 3 and 5 vol.% titanium-coated diamond/copper composites; (**b**) the corresponding processing window of the laser power and scanning rate; (**c**) The SLM manufactured morphology: top and front view of the 1, 3 and 5 vol.% titanium-coated diamond/copper composite, respectively.

**Figure 4 micromachines-13-00724-f004:**
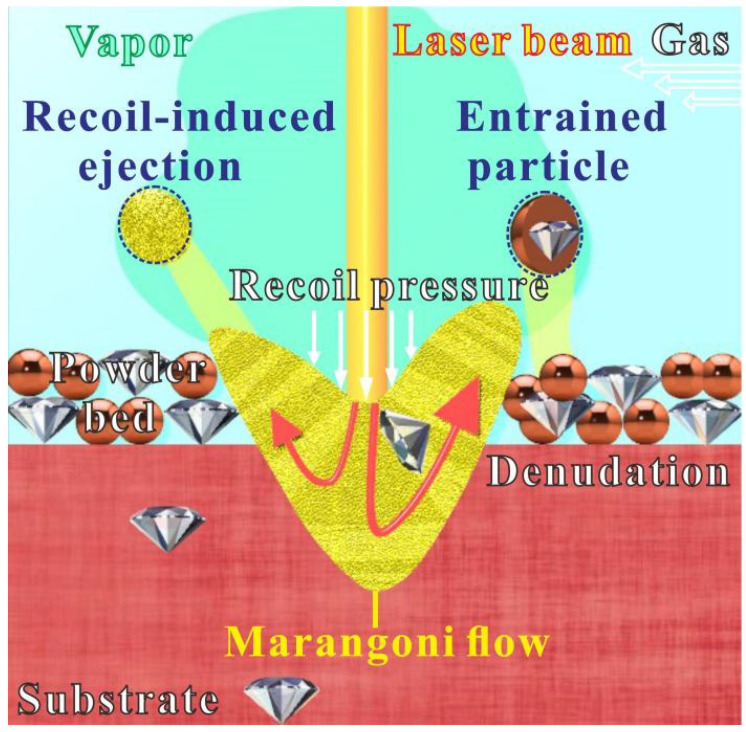
Spatter ejection phenomenon in SLM of the coated diamond/copper combined materials.

**Figure 5 micromachines-13-00724-f005:**
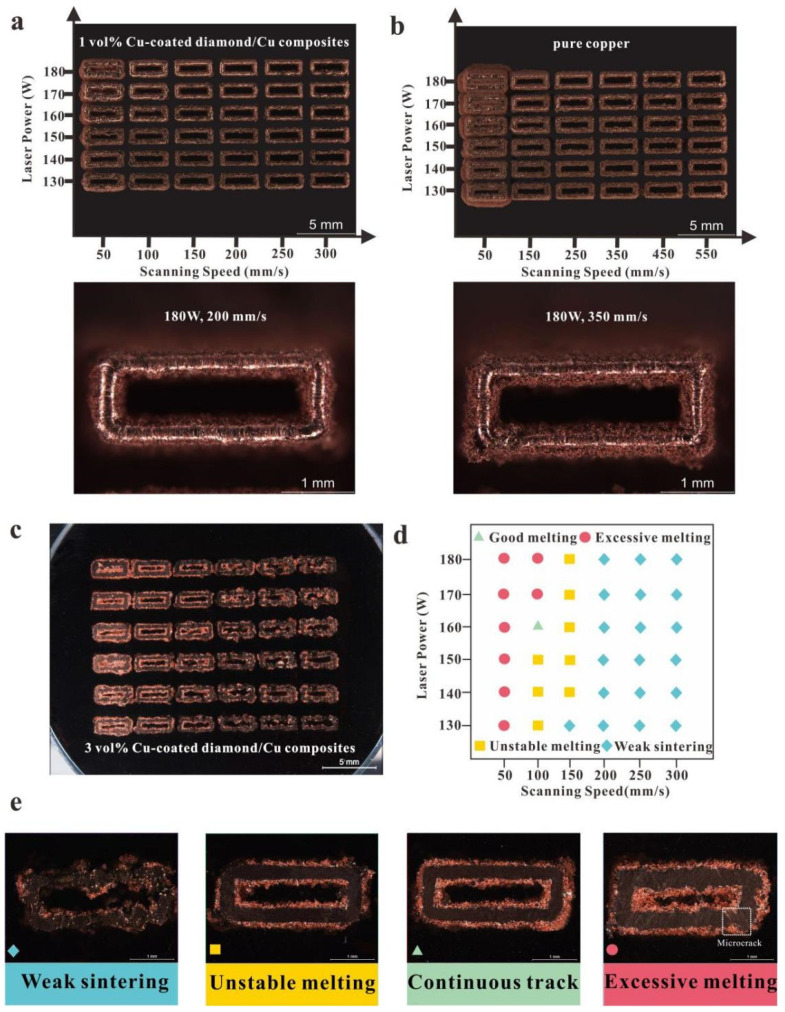
1, 3 vol.% copper-coated diamond/copper combined materials and pure copper featuring various process parameters within the XY plane and surface morphologies: (**a**) 1 vol.% copper-coated diamond/copper composites; (**b**) Pure copper; (**c**,**d**) 3 vol.% copper-coated diamond composites and process window of laser power and scanning rate; (**e**) Typical track types of zones A, B, C, and D.

**Figure 6 micromachines-13-00724-f006:**
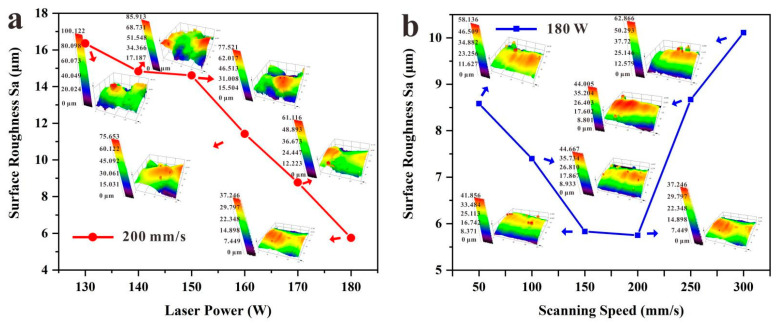
1 vol.% copper-coated diamond/copper combined materials relationship between the surface roughness and (**a**) laser power and (**b**) scanning rate.

**Figure 7 micromachines-13-00724-f007:**
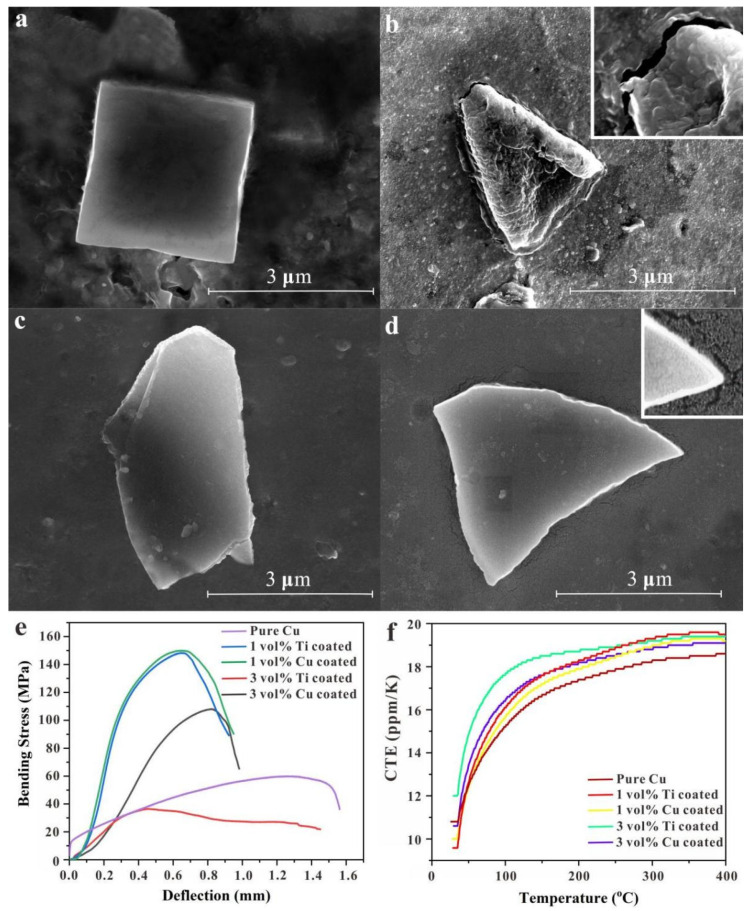
SEM images of the copper matrix and diamond bonding of (**a**) 1 vol.% and (**b**) 3 vol.% titanium-coated diamond/copper combined materials; (**c**) 1 vol.% and (**d**) 3 vol.% copper-coated diamond/copper combined materials; (**e**) The bending stress and (**f**) CTE values of titanium and copper-coated diamond/copper combined materials and pure copper.

**Table 1 micromachines-13-00724-t001:** The coated diamond/copper composite compositions.

Diamond vol.%	Diamond wt.%	Total Mass (g)	Coated Diamond Quality (g)	Copper Quality (g)
1	0.40	50	0.20	49.80
3	1.20	50	0.60	49.40
5	2.03	50	1.01	48.99

## Data Availability

Not applicable.

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
