# Peer review of "Fabrication of Titanium and Copper-Coated Diamond/Copper Composites via Selective Laser Melting"

_micromachines, 2022, doi:10.3390/mi13050724_

Round 1

Reviewer 1 Report

Dear Authors,

You have clearly focused on optimising the printing parameters and developing new approaches for the further construction of a wider range of diamond-particle-reinforced metal matrix composites in the paper, which is heavily focused on titanium and copper-coated diamond/copper composites via selective laser melting. However, the optimization strategy for identifying the most affecting process parameters and the contribution of each and every process parameter to producing the above requirement is absolutely missing.

Reviewer 2 Report

This work perfectly suits the scope of this Journal. It deserves to be published. I suggest only some corrections of language, and misprints (subscripts in chemical formulas,...)

Reviewer 3 Report

The article is devoted to the study of the properties of composites based on diamond/copper obtained by laser sintering. Undoubtedly, the results presented by the authors are of high scientific novelty and practical significance, and are also promising for practical research. In general, the presented results of the study can be accepted for publication after the authors provide answers to all the questions raised by the reviewer during the reading of the article.

1. In the abstract, the authors need to more clearly state the purpose and relevance of this work.
2. There is a lot of information about the reinforcing properties of diamonds, the authors should explain what caused the choice of the concentration of diamonds in such proportions.
3. The scheme for obtaining structures is quite illustrative and fully reflects the proposed technique; however, the procedure for annealing and subsequent laser sintering is not entirely clear. Authors should explain these methods.
4. Exactly how the degree of roughness changes after laser sintering.
5. Authors should compare their results with other similar composites to determine the effectiveness of their proposed method.
6. Conclusion requires significant revision.

Round 2

Reviewer 1 Report

Dear Authors,

Your response is appreciated, and I recommend that in your future work, you concentrate on optimising the process parameters.